# New Imadazopyrazines with CDK9 Inhibitory Activity as Anticancer and Antiviral: Synthesis, In Silico, and In Vitro Evaluation Approaches

**DOI:** 10.3390/ph16071018

**Published:** 2023-07-18

**Authors:** Aisha A. Alsfouk, Hanan M. Alshibl, Najla A. Altwaijry, Ashwag Alanazi, Omkulthom AlKamaly, Ahlam Sultan, Bshra A. Alsfouk

**Affiliations:** 1Department of Pharmaceutical Sciences, College of Pharmacy, Princess Nourah bint Abdulrahman University, P.O. Box 84428, Riyadh 11671, Saudi Arabia; aaalsfouk@pnu.edu.sa (A.A.A.); naaltwaijry@pnu.edu.sa (N.A.A.); asalanzi@pnu.edu.sa (A.A.); omalkmali@pnu.edu.sa (O.A.); ahmsultan@pnu.edu.sa (A.S.); 2Department of Pharmaceutical Chemistry, College of Pharmacy, King Saud University, P.O. Box 2457, Riyadh 11451, Saudi Arabia; halshibl@ksu.edu.sa

**Keywords:** cyclin-dependent kinase, kinase inhibitor, COVID-19, 229E, coronavirus

## Abstract

This study describes the synthesis and biological activity of new imadazopyrazines as first-in-class CDK9 inhibitors. The inhibition of CDK9 is a well-established therapeutic target in cancer therapy. The new compounds were assessed using an in vitro kinase assay against CDK9. In this assay, compound **1d** exhibited the highest CDK9 inhibition with an IC_50_ of 0.18 µM. The cytotoxicity effect of the novel compounds was evaluated in three cancer cell lines: HCT116, K652, and MCF7. The results of this assay showed a correlation between the antiproliferative effect of the inhibitors and their CDK9 inhibitory effect in the biochemical assay. This suggests CDK9 inhibition as a mechanistic pathway for their anticancer effect. Several compounds demonstrated potent cytotoxic effects with single-digit micromolar IC_50_ values yielded through an MTT assay. The compounds with the most promising data were further assessed for their antiviral activity against human Coronavirus 229E. The results showed that compound **4a** showed the highest antiviral potency with an IC_50_ of 63.28 µM and a selectivity index of 4.8. In silico target prediction data showed that **4a** displayed a good affinity to proteases. The result of the docking studies of **4a** with COVID-19 main protease revealed a high binding affinity, which confirmed the results obtained from in vitro study. The physiochemical and in silico pharmacokinetic parameters indicated reasonable drug-likeness properties of the new compounds, including solubility, lipophilicity, absorption, oral bioavailability, and metabolic stability. Further lead optimization of this novel scaffold could lead to a revolution of a new class of preclinical CDK9 agents.

## 1. Introduction

Cyclin-dependent kinases are an integral component of the cell cycle machinery. They are a family of serine/threonine kinases that are dimerized with cyclins to catalyze the phosphorylation of various endogenous substrates that result in cell-cycle progression at different phases [1,2,3]. CDK9 is a member of this family that, in collaboration with cyclin T, forms positive transaction elongation factor b (p-TEFB), which plays an important role in RNA transcription [4,5].

There is satisfactory evidence that supports the role of CDK9 in cancer via controlling proapoptotic proteins and the promotion of cell proliferation [6,7,8]. Its pathogenic function has been well-established in a number of malignancies such as breast, pancreatic, prostate, lymphomas, and others [6,9,10,11,12,13,14].

Since the recognition of CDK9 as a therapeutic possibility for treating cancer, large pharmaceutical companies have developed several scaffolds as CDK9 inhibitors. The majority of them were developed as ATP-competitive inhibitors with small molecular weights and good drug-likeness properties. Several of them were successfully progressed to clinical trials as antiproliferative agents targeting various types of cancers [2,9,15,16,17,18,19,20,21,22,23]. The initial clinical experience with the inhibition of CDK9 was with nonselective CDK9 inhibitors such as flavopiridol, dinaciclib, and SNS-032 (Table 1). The clinical results with these inhibitors showed good responses, but they also showed a high incidence of adverse effects [24,25,26,27,28,29,30]. Currently, there are three highly selective CDK9 inhibitors in clinical evaluation targeting hematological malignancies: atuveciclib, BAY-1251152, and AZD4573 (Table 2) [31,32,33] 

In 2019, an imadazopyrazine **1** (Figure 1) was isolated from natural resources and identified by virtual screening as a first-in-class CDK9 inhibitor with an IC_50_ of 7.88 µM, and it exhibited an antiproliferative effect against a panel of human breast cancer cell lines [38,39]. Further optimization of this novel scaffold was conducted by our team with the identification of 2-phenylimidazopyrazin-3-amine **2** as a CDK9 inhibitor with submicromolar IC_50_ values and a potent antiproliferative effect against a panel of cancer cell lines with single-digit IC_50_ [40]. The current study aims to further develop the novel imidazopyrazine scaffold targeting CDK9 in cancer. The activity of the new compounds has been evaluated using an MTT assay against three cancer cell lines, the binding mode of the novel compounds has been studied in silico, and their pharmacokinetics and drug-likeness properties have been examined. In addition, based on the previous data, this scaffold possesses some antiviral activity against the human Coronavirus [40,41]. Therefore, the antiviral effect of the agents has been assessed in this study. 

## 2. Results and Discussion 

### 2.1. Chemistry

#### 2.1.1. Synthetic Approach

The synthetic route used to synthesize the new compounds in this work, for **1a**–**4d**, is illustrated in Figure 1. The novel compounds were synthesized using a Groebke–Blackburn–Bienaymé reaction; it is a one-pot multicomponent reaction where 2-aminopyrazine, aldehydes, and isocyanides were reacted in the presence of scandium (III)-triflate catalyst and a solvent mixture of dichloromethane and methanol (3:1). This reaction was conducted in a microwave using a temperature of 150° for 30 min. The purification of the final compounds was carried out using column chromatography that utilized the mobile phases of ethyl acetate and hexane. 

Compounds **1a**–**1d** carry furan-3-yl in position 2 of imadazopyrazine and different amines in position 3, such as *t*-butyl, cyclohexyl, benzyl, and 4-methoxyphenyl. These compounds were obtained in a good yield of 86–91%. Compounds **2a**–**2d** carry phenyl-2,4 diol in position 2 of imadazopyrazine, and they were obtained in a good yield of 79–96%. Compounds **3a**–**3d** have 4-(dimethylamino)-pyridin-3-yl in position 2 of imadazopyrazine. The yield of this series ranged from 43 to 88%. Compounds **4a**–**4d** have 2-fluoro- pyridin-4-yl in position 2 of imadazopyrazine. The yield of these compounds was 79–97%. 

The construction of the imadazopyrazine core was established using (Nuclear Magnetic Resonance (NMR). In the ^1^H NMR spectrum of all compounds, three aromatic signals appeared that corresponded to the methylene protons of the pyrazine ring. For instance, the ^1^H NMR spectrum of **3a** displayed the presence of two pair of doublets at 7.79 (d, *J* = 4.12 Hz, 1 H) and 8.34 (d, *J* = 4.12 Hz, 1 H) in addition to the presence of singlet signal at 8.84 (s, 1 H). These three protons represent imadazopyrazine’s methylene protons (Figure 2).

#### 2.1.2. Rational of Molecule Design

A previous study by our team demonstrated that compounds with lipophilic substituents at position 3 of imadazopyrazine have optimum activity (compounds **1** and **2** in Figure 1) [40]. Moreover, docking studies showed that two to three carbons is the optimal distance between the lipophilic substituent and the imadazopyrazine core to occupy the G-rich pocket and form several hydrophobic interactions with Ile25, Gly26, and Val33 (Figure 3). Therefore, phenyl, benzyl, cyclohexyl, and t-butyl groups were selected to be tested at this position (R in Figure 1). The data from a previous study showed that compounds with 4-pyridinyl and 4-hydroxyphenyl in position 2 of imadazopyrazine have the optimum activity (compounds **1** and **2** in Figure 1). Also, the docking studies showed that the hydrogen bond donor/acceptor group of 4-pyridinyl and 4-hydroxyphenyl is pointed toward the solvent-exposed area and forms a hydrogen bond with Asp109. Therefore, several heterocycles with hydrogen bond donor/acceptor moieties were incorporated in 2-imadazopyrazine.

### 2.2. Anticancer Activity 

#### 2.2.1. CDK9 Activity 

CDK9 is a member of the kinase family of enzymes that has a role in cell growth through the activation of RNA polymerase II via phosphorylation [4,5]. Its pathogenic function has been well-established in a number of malignancies such as acute myeloid leukemia, as well as pancreatic and prostate cancers. Its pathogenetic effects are mediated through the regulation of antiapoptotic proteins that are essential for tumor initiation and progression [6,9,10,11,12,13].

In this work, the synthesized compounds were assessed for their inhibition against an isolated CDK9 enzyme using the biochemical kinase assay. Standard dinaciclib (a well-established potent CDKs inhibitor) was used as a control in this experiment. 

Table 3 displays the outcomes of the kinase assay. The data showed that these compounds revealed good CDK9 inhibition with an IC_50_ value of 0.18–1.78 µM. Compounds with furan-3-yl in position 2 of imadazopyrazine exhibited the highest CDK9 inhibitory activity with sub-µM IC_50_ values. In particular, **1d**, a compound with a 4-methoxyphenyl amine in position 3 of imadazopyrazine, exhibited the most potent CDK9 inhibitory activity among all the tested compounds with an IC_50_ of 0.18 µM.

In addition, the data indicated that derivatives with *t*-butylamine in position 3 of imadazopyrazine (**1a**, **2a**, **3a**, and **4a**) showed good CDK9 inhibition with a low IC_50_ range of 0.19–0.45 µM.

#### 2.2.2. Cytotoxicity Assay 

The cytotoxicity effect of the new compounds was assessed in an MTT assay against chronic myelogenous leukemia (K652), colorectal (HCT116), and breast (MCF7) cancer cell lines. Staurosporine standard was used as a control in this experiment. The data is presented in Table 4 as IC_50_ values (in µM).

The results of the MTT assay indicated that the new compounds described in this work have good cytotoxicity effects, with IC_50_ values in the three cell lines ranging from 10.65 to 143.79 µM. Furthermore, the cytotoxic effects of the inhibitors in this assay against several cell lines were allied to their CDK9 inhibitory effects in the biochemical assay. The derivatives with the highest potency in the CDK9 primary assay showed the most potent cytotoxic effects in the MTT assay (for example **1d**, **1a**, **3b**, and **4a** exhibited IC_50_ values of 0.18, 0.19, 0.23, and 0.24 µM in the biochemical assay, respectively, as well as average IC_50_ values of 11.62, 12.61, 10.65, and 20.73 µM in the MTT assay, respectively). In addition, the agents that exhibited the weakest CDK9 inhibition in the primary assay such as **1b**, **3d**, and **4c** (with IC_50_ values of 1.78, 1.66, and 1.11 µM, respectively) showed the weakest cytotoxic effects in the antiproliferative assay (average IC_50_ values of 143.79, 140.04, and 92.79 µM, respectively), thus suggesting the CDK9 inhibition as a mechanistic pathway of their anticancer effects.

To ensure that the cytotoxicity effect of our compounds observed on the previous assay was selective to cancer cells, the most cytotoxic agents to cancer cells (**1a**, **1c**, **1d**, **3b**, and **4b**) were evaluated on normal cells. The data is presented in Table 5; our compounds exhibited weak cytotoxicity against normal noncancerous FHC cells, with IC_50_ values ranging from 58.64–187.31 µM, as well as a selectivity index ranging from 2.5–20.5. This data indicated that the most promising agents showed acceptable safety for the normal cells. 

#### 2.2.3. Molecular Docking Study into CDK9 Active Site 

Simulation docking studies into the ATP binding site of the CDK9 enzyme of compounds **1d**, **3b**, **3c**, and **4a** were conducted using AutoDock Vina embedded in PyRx 0.8 software. In these studies, the CDK9 complex with a UT5 ligand was used as a macromolecule (PDB: 7NWK). The validity of the docking studies was carried out initially by the redocking of a cocrystalized inhibitor (UT5). It showed a docking score of −8.5 kcal/mol with an RMSD value of 1.6 Å. The main hydrogen bond interactions between the imdazopyridine ring of UT5 and the Cys106 residue of the protein were obtained.

In general, the tested compounds fitted well into the ATP binding site and showed several interactions with key amino acid residues in the catalytic site of the CDK9 enzyme (Figure 3, Table 6). All tested compounds showed interactions between the hinge region of the CDK9 enzyme and imadazopyrazine by forming hydrogen bonds with Cys106 and Asp104. 

In compounds **3b**, **3c**, and **4a**, the lipophilic side chain of the alkylamine or benzylamine in position 3 of imadazopyrazine formed several hydrophobic interactions with Ile25, Gly26, and Val33 in the G-rich loop, while the pyridine ring in position 2 of imadazopyrazine pointed toward the solvent-exposed areas and formed a hydrogen bond with Asp109 (Figure 3A,C).

Compound **1d** adapted a flipped orientation in comparison to the poses of compounds **3b**, **3c**, and **4a**, where the furan ring of **1d** in position 2 of the imadazopyrazine occupied a region down the G-loop and interacted with Gly28, Gly26, and Ile25, while the methoxyphenyl ring in position 3 of the imadazopyrazine engaged the hydrophobic pocket and interacted with Phe103, Val166, and Val79 (Figure 3B).

In terms of biological activity, **1d** demonstrated the most potent activity among all the tested compounds in both the CDK9 and cytotoxicity assays (Section 2.2.1 and Section 2.2.2). This may indicate that the lipophilic small group in position 2 of the imadazopyrazine is more favorable than the ionizable groups (such as pyridine and phenol). The docking studies showed that **1d** has a flipped orientation in comparison to the poses of compounds **3b**, **3c**, and **4a** (Figure 3). This flipped orientation, where the 2-imadazopyrazine substituent occupies a region down the G-loop and 3-imadazopyrazine moieties engaged in the hydrophobic pocket, seems to be more favorable for biological activity.

### 2.3. Antiviral Activity 

#### 2.3.1. 229E Inhibitory Assay

The antiviral effect of the compounds was examined in a human Coronavirus (HCoV-229E) inhibitory assay. The standard ribavirin (a well-established antiviral agent) was used as a positive control in this experiment. 

The outcomes of this assay are displayed in Table 7. The data showed that compound **4a** showed good antiviral activity. In comparison to the standard antiviral agent ribavirin, **4a** demonstrated better antiviral activity against HCoV-229E with an IC_50_ of 63.28 µM. It also showed a weak cytotoxicity effect on the target cells at the concentrations that achieved its anticoronaviral effect (0.1–1000 µg/mL), with a 50% cellular cytotoxicity concentration (CC_50_) of 303.15 µM and a selectivity index (SI) of 4.8 [42,43]. 

#### 2.3.2. Antiviral Target Prediction and Molecular Docking Studies 

A target prediction study was performed in silico using SwissTarget [44]. Figure 4 shows the result of the target prediction of compound **4a**, which revealed a good affinity to protease enzymes. This suggests the inhibition of protases enzyme as a mechanistic pathway of the observed anticoronaviral activity of **4a** in an in vitro assay.

Docking studies of the compound **4a** with a COVID-19 main protease were conducted using AutoDock Vina embedded in PyRx 0.8 software. HCoV-229E is an isoform of the Coronavirus that shows high homologous sequence similarity to the SARS-CoV-2 isoform; the virus caused the pandemic respiratory disease in 2019 (COVID-19) [45,46]. Therefore, the main protease was selected as a potential target for studying the binding mode of the novel antiviral agents in this work. In these studies, the main protease of COVID-19 in a complex with X77 was used as a macromolecule (PDB: 6W63). The validity test of our docking analysis was conducted initially through redocking of the X77 inhibitor, which was the cocrystalized ligand. It showed a docking score of −9.7 kcal/mol with an RMSD value of 1.7 Å. The main H-bonding between the Gly143 and Glu166 amino acids of the receptor and the cocrystalized X77 were gained.

The docking outcomes showed that the H-bonding between the pyridine ring of compound **4a** formed with the Glu166 amino acid residue. It also revealed a hydrogen bond interaction between the amino group of the imadazopyrazine at position 3 of **4a** and the Phe140 residue. The imadazopyrazine formed several hydrophobic interactions with Cy145, His41, and Met165. The fluorine atom at position 2 of the pyridine ring interacted with Asn142 (Figure 5, Table 8).

### 2.4. In Silico Prediction of Drug-Likeness Properties 

#### 2.4.1. Molecular Structured and Physicochemical Properties 

The physiochemical properties of the selected inhibitors (**1a**, **1d**, **2c**, **3b**, **4a**, and **4b**) were estimated using SwissADME [47]. All the studied compounds displayed good physiochemical properties, as shown in Table 9. All the compounds were shown to be in line with Lipinski’s rule of five, with zero violations, and are therefore expected to be orally bioavailable. Their molecular weights were shown to be <500, and they demonstrated optimal lipophilicity (Log *P* values ranged from 1–5), optimal polarity (tPSA values < 90 Å^2^), and reasonable aqueous solubility (Log S > −4 mol/L). 

The compounds with furan-3-yl in position 2 of the imadazopyrazine displayed the highest lipophilicity. In the biochemical and cell assays, they also exhibited the most potent activity among all the tested compounds. This could indicate that the lipophilic small group is more favorable than the ionizable groups (such as pyridine and phenol) in position 2 of the imadazopyrazine. 

#### 2.4.2. ADMET Studies 

In silico pharmacokinetic studies of the inhibitors **1d**, **3b**, and **4a** were conducted through the ADMETLab platform [48]. Table 10 shows the data of these studies. The three studied compounds displayed high oral absorption by demonstrating good intestinal permeability and low efflux liability (with a Papp index > −5.15, which is a positive test result for human intestinal absorption). None of these compounds are predicted to be a P-glycoprotein substrate or inhibitor (except for **1d**, which is expected to be an inhibitor of P-glycoprotein). All three compounds showed good bioavailability scores. With respect to distribution, the volume of distribution of the studied compounds was within the optimal range (0.04–20 L/kg), and plasm protein binding (PPB) was within the acceptable level (<90%). The three tested derivatives are expected to cross the blood–brain barrier (BBB). Regarding the metabolism, all three compounds are estimated to be an inhibitor of CYP3A4 (the main metabolizing CYP450 isoform), but none of them are expected to be metabolized by the same isoform. With respect to excretion, the compounds displayed half-lives > 0.5 h and a clearance that was <15 mL/min/kg, which are considered acceptable values in the drug development process. With respect to the toxicity profile, all three of the studied derivatives were expected to block the hERG channel and produce mutagenicity, as well as liver injury, but none of them were shown to be a skin sensitizer. 

## 3. Materials and Methods

### 3.1. Chemistry 

All starting materials, building blocks, catalysts, solvents, and reagents, including 2-aminopyrazine, isocyanides, aldehydes, scandium (III)-trifluoromethanesulfonate, methanol, and dichloromethane, were purchased from Sigma-Aldrich and TCI chemical companies and used directly in the reactions without prior purification. Electrothermal melting point apparatus was used to measure the melting points of the compounds without correction. The ^1^H NMR spectra at 700.17 MHz and ^13^C spectra at 176.08 MHz were obtained using Bruker Ascend 700 NMR spectrometer (Fällanden, Switzerland) and DMSO-*d_6_* as solvents in all samples. TLC was used to monitor the progression of the reactions using aluminum sheets of precoated silica gel (60 F254, Merck) and visualized under UV light at 365 and 254.

General procedure: 

In a microwave vial, a mixture of 2-aminopyrazine (0.52 mmol), required isocyanide (0.52 mmol), required aldehyde (0.4 mmol), scandium (III)-trifluoromethanesulfonate catalyst (0.015 mmol), and 2 mL of a solvent system of 3-to-1 dichloromethane-to-methanol were added. The vial was then sealed and heated to 150 °C for 30 min in the microwave. The reaction mixture was then allowed to cool down to room temperature and concentrated. The reaction residue was then extracted with 5 mL dichloromethane three times. The organic extract was then collected, dried over magnesium sulfate, and evaporated. The residue was then purified using column chromatography using hexane and ethyl acetate. 

*The N-(Tert-butyl)-2-(furan-3-yl) imidazo[1,2-a] pyrazin-3-amine* (**1a**) *Yield*: 89.3%; yellow oil; ^1^H NMR (700 MHz, DMSO-*d*_6_) δ ppm 1.09 (br. s., 9 H), 4.76 (s, 1 H), 7.13 (s, 1 H), 7.76 (br. s., 1 H), 7.84 (br. s., 1 H), 8.32 (s, 1 H), 8.38 (br. s., 1 H), 8.89 (s, 1 H); ^13^C NMR (176 MHz, DMSO-*d*_6_) δ ppm 30.53, 56.85, 110.46, 117.60, 120.40, 125.57, 128.81, 135.55, 137.31, 141.81, 142.52, and 143.76. *m/z* (ESI-MS) [M]^+^ 257.13.*The N-Cyclohexyl-2-(furan-3-yl) imidazo[1,2-a] pyrazin-3-amine* (**1b**) *yield:* 86.4%; yellow oil; ^1^H NMR (700 MHz, DMSO-*d*_6_) δ ppm 1.09 (br. s., 9 H), 1.10–1.15 (m, 3 H), 1.28–1.33 (m, 2 H), 1.53 (br. s., 1 H), 1.66 (d, *J* = 11.83 Hz, 2 H), 1.75 (d, *J* = 12.69 Hz, 2 H), 2.86–2.90 (m, 1 H), 4.92 (d, *J* = 7.10 Hz, 1 H), 7.79 (s, 1 H), 7.85 (d, *J* = 4.52 Hz, 1 H), 7.87 (s, 1 H) 8.25 (s, 1 H), 8.30 (dd, *J* = 4.52, 1.08 Hz, 1 H), 8.88 (s, 1 H); ^13^C NMR (176 MHz, DMSO-*d*_6_) δ ppm 25.07, 25.81, 34.19, 56.82, 109.78, 116.72, 120.13, 127.17, 120.95, 132.89, 136.65, 141.06, 142.46, and 144.16. *m/z* (ESI-MS) [M]^+^ 283.07.*The N-Benzyl-2-(furan-3-yl) imidazo[1,2-a]pyrazin-3-amine* (**1c**) *yield:* 91.0%; yellow oil; ^1^H NMR (700 MHz, DMSO-*d*_6_) δ ppm 4.17 (d, *J* = 6.24 Hz, 2 H), 5.57 (t, *J* = 6.35 Hz, 1 H), 7.04 (s, 1 H), 7.23–7.31 (m, 5 H), 7.75 (d, *J* = 4.52 Hz, 1 H), 7.79 (s, 1 H), 8.12 (d, *J* = 4.30 Hz, 1 H), 8.20 (s, 1 H), 8.86 (s, 1 H); ^13^C NMR (176 MHz, DMSO-*d*_6_) δ ppm 51.09, 109.79, 116.46, 119.94, 127.63, 126.63, 126.75, 131.98, 132.05, 132.89, 136.47, 140.13, 141.06, 142.45, and 144.20. *m/z* (ESI-MS) [M]^+^ 291.14.*The 2-(Furan-3-yl)-N-(4-methoxyphenyl) imidazo[1,2-a] pyrazin-3-amine* (**1d**) *yield:* 88.4%; yellow oil; ^1^H NMR (700 MHz, DMSO-*d*_6_) δ ppm 3.64 (s, 3 H), 6.47 (d, *J* = 7.31 Hz, 2 H), 6.77 (d, *J* = 7.74 Hz, 2 H), 6.93 (s, 1 H), 7.75 (br. s., 1 H), 7.89 (br. s., 1 H), 7.99 (br. s., 1 H), 8.03 (br. s., 1 H), 8.04 (br. s., 1 H), 9.04 (s, 1 H); ^13^C NMR (176 MHz, DMSO-*d*_6_) δ ppm 55.69, 109.55, 114.76, 115.47, 132.04, 132.92, 134.91, 137.66, 138.87, 141.47, 142.31, 142.85, 144.41, 153.12, and 156.53. *m/z* (ESI-MS) [M]^+^ 291.14.*The 4-(3-(Tert-butylamino) imidazo[1,2-a]pyrazin-2-yl)benzene-1,3-diol* (**2a**) *yield:* 96.6%; white solid (MP: 174–176 °C); ^1^H NMR (700 MHz, DMSO-*d*_6_) δ ppm 1.00 (s, 9 H), 5.35 (s, 1 H), 6.28 (d, *J* = 2.75 Hz, 1 H), 6.34 (d, *J* = 2.06 Hz, 1 H), 7.86 (d, *J* = 4.81 Hz, 1 H), 7.89 (d, *J* = 8.94 Hz, 1 H), 8.39 (dd, *J* = 4.81, 1.37 Hz, 1 H), 8.91 (d, *J* = 1.37 Hz, 1 H), 9.51 (br. s., 1 H), 11.53 (br. s., 1 H); ^13^C NMR (176 MHz, DMSO-*d*_6_) δ ppm 29.77, 109.8, 116.7, 119.80, 127.61, 126.67, 126.79, 132.01, 132.70, 132.89, 136.54, 140.15, 141.67, 142.45, and 144.22. *m/z* (ESI-MS) [M]^−^ 297.04.*The 4-(3-(Cyclohexylamino) imidazo[1,2-a] pyrazin-2-yl)benzene-1,3-diol* (**2b**) *yield:* 79.5%; white solid (MP: 245–247 °C); ^1^H NMR (700 MHz, DMSO-*d*_6_) δ ppm 1.02–1.11 (m, 4 H), 1.20 (d, *J* = 11.00 Hz, 2 H), 1.58–1.61 (m, 2 H), 1.66 (d, *J* = 12.37 Hz, 2 H), 2.81 (tdt, *J* = 10.35, 10.35, 6.96, 3.69, 3.69 Hz, 1 H), 5.21 (d, *J* = 6.87 Hz, 1 H), 6.28 (d, *J* = 2.75 Hz, 1 H), 6.34 (dd, *J* = 8.59, 2.41 Hz, 1 H), 7.89 (d, *J* = 4.81 Hz, 1 H), 7.99 (d, *J* = 8.25 Hz, 1 H), 8.34 (dd, *J* = 4.12, 1.37 Hz, 1 H), 8.92 (d, *J* = 1.38 Hz, 1 H), 9.57 (s, 1 H), 12.15 (s, 1 H); ^13^C NMR (176 MHz, DMSO-*d*_6_) δ ppm 51.09, 109.79, 116.46, 119.94, 127.63, 126.63, 126.75, 131.98, 132.05, 132.89, 136.47, 140.13, 141.06, 142.45, and 144.20. *m/z* (ESI-MS) [M]^+^ 325.17.*The 4-(3-(Benzylamino)imidazo[1,2-a]pyrazin-2-yl)benzene-1,3-diol* (**2c**) *yield:* 91.8%; white solid (MP: 167–169 °C); ^1^H NMR (700 MHz, DMSO-*d*_6_) δ ppm 4.06 (d, *J* = 6.87 Hz, 2 H), 5.65 (t, *J* = 6.19 Hz, 1 H), 6.31 (d, *J* = 2.06 Hz, 1 H), 6.33–6.36 (m, 1 H), 7.21–7.23 (m, 4 H), 7.29–7.32 (m, 1 H), 7.79 (d, *J* = 4.12 Hz, 1 H), 7.98 (d, *J* = 8.94 Hz, 1 H), 8.14 (dd, *J* = 4.12, 1.37 Hz, 1 H), 8.89 (d, *J* = 1.37 Hz, 1 H), 9.60 (s, 1 H), 12.10 (s, 1 H)); ^13^C NMR (176 MHz, DMSO-*d*_6_) δ ppm 51.29, 103.64, 107.74, 109.38, 116.25, 126.32, 127.78, 128.78, 128.83, 129.32, 129.51, 134.65, 136.85, 139.77, 141.63, 158.34, and 159.44. *m/z* (ESI-MS) [M]^+^ 331.06.*The 4-(3-((4-Methoxyphenyl) amino)imidazo[1,2-a]pyrazin-2-yl)benzene-1,3-diol* (**2d**) *yield:* 79.9%; white solid (MP: 186–188 °C); ^1^H NMR (700 MHz, DMSO-*d*_6_) δ ppm 3.60 (s, 3 H), 6.20–6.22 (m, 1 H), 6.29 (d, *J* = 2.75 Hz, 1 H), 6.44 (m, *J* = 8.94 Hz, 2 H), 6.72 (m, *J* = 8.94 Hz, 2 H), 7.75 (d, *J* = 8.93 Hz, 1 H), 7.93 (d, *J* = 4.81 Hz, 1 H), 8.00 (s, 1 H) 8.02 (dd, *J* = 4.12, 1.37 Hz, 1 H), 9.09 (s, 1 H), 9.64 (s, 1 H), 12.49 (s, 1 H); ^13^C NMR (176 MHz, DMSO-*d*_6_) δ ppm 55.74, 103.63, 107.75, 107.96, 114.81, 115.58, 116.44, 119.63, 128.70, 130.62, 135.32, 138.60, 139.75, 141.89, 153.26, 159.15, and 159.81. *m/z* (ESI-MS) [M]^+^ 349.12.*The N-(Tert-butyl)-2-(6-(dimethylamino) pyridin-3-yl)imidazo[1,2-a]pyrazin-3-amine* (**3a**) *yield:* 98%; white solid (MP: 126–128 °C); ^1^H NMR (700 MHz, DMSO-*d*_6_) δ ppm 0.97 (s, 9 H), 3.02 (s, 6 H), 4.72 (s, 1 H), 6.67 (d, *J* = 8.94 Hz, 1 H), 7.79 (d, *J* = 4.12 Hz, 1 H), 8.22 (d, *J* = 8.93 Hz, 1 H), 8.34 (d, *J* = 4.12 Hz, 1 H), 8.84 (s, 1 H), 8.84 (s, 1H); ^13^C NMR (176 MHz, DMSO-*d*_6_) δ ppm 30.52, 38.16, 56.66, 105.57, 117.54, 118.36, 124.96, 128.85, 136.98, 137.20, 139.86, 142.22, 147.54, and 158.65. *m/z* (ESI-MS) [M]^+^ 311.20.*The N-Cyclohexyl-2-(6-(dimethylamino) pyridin-3-yl)imidazo[1,2-a]pyrazin-3-amine* (**3b**) *yield:* 85.1%; white solid (MP: 191–193 °C); ^1^H NMR (700 MHz, DMSO-*d*_6_) δ ppm 1.04 (br. s., 3 H), 1.19–1.24 (m, 2 H), 1.45 (br. s., 1 H), 1.58 (br. s., 2 H), 1.65 (d, *J* = 12.37 Hz, 2 H), 2.79 (dd, *J* = 10.31, 3.44 Hz, 1 H), 3.03 (s, 6 H), 4.91 (d, *J* = 6.87 Hz, 1 H), 6.70 (d, *J* = 8.94 Hz, 1 H), 7.78 (d, *J* = 4.81 Hz, 1 H), 8.23 (dd, *J* = 8.94, 2.75 Hz, 1 H), 8.26 (d, *J* = 4.12 Hz, 1 H), 8.81 (s, 1 H), 8.86 (s, 1 H); ^13^C NMR (176 MHz, DMSO-*d*_6_) δ ppm 25.02, 25.84, 34.07, 38.19, 56.84, 106.01, 116.66, 117.90, 118.35, 126.36, 128.99, 136.00, 136.59, 142.14, 146.69, and 158.64. *m/z* (ESI-MS) [M]^+^ 336.96.*The N-Benzyl-2-(6-(dimethylamino) pyridin-3-yl)imidazo[1,2-a]pyrazin-3-amine* (**3c**) *yield:* 43.3%; yellow oil; ^1^H NMR (700 MHz, DMSO-*d*_6_) δ ppm 3.05 (s, 6 H), 4.08 (d, *J* = 6.87 Hz, 2 H), 5.58 (t, *J* = 6.53 Hz, 1 H), 6.71 (d, *J* = 8.94 Hz, 1 H), 7.20–7.22 (m, 4 H), 7.68 (d, *J* = 4.81 Hz, 1 H), 7.76–7.88 (m, 1 H), 8.09 (dd, *J* = 4.81, 1.37 Hz, 1 H), 8.17 (dd, *J* = 8.94, 2.75 Hz, 1 H), 8.79 (d, *J* = 1.37 Hz, 1 H), 8.82 (d, *J* = 2.06 Hz, 1 H); ^13^C NMR (176 MHz, DMSO-*d*_6_) δ ppm 38.22, 51.22, 106.07, 106.44, 116.36, 117.94, 121.71, 127.67, 128.61, 128.78, 136.12, 136.42, 140.10, 142.22, 146.68, 158.64, and 189.82. *m/z* (ESI-MS) [M]^+^ 345.22.*The 2-(6-(Dimethylamino) pyridin-3-yl)-N-(4-methoxyphenyl) imidazo[1,2-a]pyrazin-3-amine* (**3d**) *yield:* 47.7%; white solid (MP: 180–182 °C); ^1^H NMR (700 MHz, DMSO-*d*_6_) δ ppm 3.00 (s, 6 H), 3.59 (s, 3 H), 6.40 (m, *J* = 8.94 Hz, 2 H), 6.67 (d, *J* = 8.94 Hz, 1 H), 6.72 (m, *J* = 8.94 Hz, 2 H), 7.82 (d, *J* = 4.12 Hz, 1 H), 7.96 (dd, *J* = 4.81, 1.37 Hz, 1 H), 7.97 (s, 1 H), 8.09 (dd, *J* = 8.94, 2.06 Hz, 1 H), 8.69 (d, *J* = 2.06 Hz, 1 H), 8.98 (d, *J* = 1.37 Hz, 1 H); ^13^C NMR (176 MHz, DMSO-*d*_6_) δ ppm 38.14, 55.75, 106.11, 114.59, 115.61, 116.60, 117.11, 120.28, 122.57, 129.61, 135.88, 137.70, 139.10, 142.66, 146.80, 135.11, and 158.87. *m/z* (ESI-MS) [M]^+^ 361.31.*The N-(Tert-butyl)-2-(2-fluoropyridin-4-yl) imidazo[1,2-a] pyrazin-3-amine* (**4a**) *yield:* 87.7%; yellow oil; ^1^H NMR (700 MHz, DMSO-*d*_6_) δ ppm 1.00 (s, 9 H), 5.04 (s, 1 H), 7.85 (s, 1 H), 7.87 (d, *J* = 4.12 Hz, 1 H), 8.11 (d, *J* = 4.81 Hz, 1 H), 8.27 (d, *J* = 4.81 Hz, 1 H), 8.43 (d, *J* = 4.81 Hz, 1 H), 8.98 (s, 1 H); ^13^C NMR (176 MHz, DMSO-*d*_6_) δ ppm 30.40, 57.26, 107.40, 118.13, 120.70, 128.34, 129.30, 136.70, 148.22, 163.34, and 164.89. *m/z* (ESI-MS) [M]^+^ 286.04.*The N-Cyclohexyl-2-(2-fluoropyridin-4-yl) imidazo[1,2-a] pyrazin-3-amine* (**4b**) *yield:* 81.2%; yellow oil; ^1^H NMR (700 MHz, DMSO-*d*_6_) δ ppm 1.04–1.10 (m, 3 H), 1.25–1.31 (m, 2 H), 1.47 (br. s., 1 H), 1.59–1.63 (m, 2 H), 1.70 (d, *J* = 12.37 Hz, 2 H), 2.79–2.86 (m, 1 H), 5.30 (d, *J* = 7.56 Hz, 1 H), 7.77 (s, 1 H), 7.86 (d, *J* = 4.81 Hz, 1 H), 8.06 (d, *J* = 5.50 Hz, 1 H), 8.29 (d, *J* = 5.50 Hz, 1 H), 8.37 (dd, *J* = 4.12, 1.37 Hz, 1 H), 8.97 (s, 1 H); ^13^C NMR (176 MHz, DMSO-*d*_6_) δ ppm 25.11, 25.76, 34.17, 57.59, 105.92, 117.29, 119.44, 129.34, 130.71, 134.14, 136.67, 144.22, 148.51, 163.64, and 165.18. *m/z* (ESI-MS) [M]^+^ 312.11.*The N-Benzyl-2-(2-fluoropyridin-4-yl) imidazo[1,2-a] pyrazin-3-amine* (**4c**) *yield:* 79.2%; yellow oil; ^1^H NMR (700 MHz, DMSO-*d*_6_) δ ppm 4.14 (d, *J* = 6.87 Hz, 2 H), 5.94 (t, *J* = 6.53 Hz, 1 H), 7.16–7.19 (m, 5 H), 7.65 (s, 1 H), 7.77 (d, *J* = 4.81 Hz, 1 H), 7.96 (d, *J* = 4.81 Hz, 1 H), 8.20 (dd, *J* = 4.47, 1.72 Hz, 1 H), 8.25 (d, *J* = 5.50 Hz, 1 H), 8.94 (d, *J* = 1.37 Hz, 1 H); ^13^C NMR (176 MHz, DMSO-*d*_6_) δ ppm 51.50, 106.11, 117.07, 119.59, 127.81, 128.70, 128.82, 129.11, 131.01, 134.15, 136.54, 139.65, 144.13, 148.38, 163.57, and 165.12. *m/z* (ESI-MS) [M]^+^ 320.11.*The 2-(2-Fluoropyridin-4-yl)-N-(4-methoxyphenyl) imidazo[1,2-a]pyrazin-3-amine* (**4d**) *yield:* 97.0%; yellow oil; ^1^H NMR (700 MHz, DMSO-*d*_6_) δ ppm 3.60 (s, 3 H), 6.48 (d, *J* = 8.94 Hz, 2 H), 6.74 (d, *J* = 8.94 Hz, 2 H), 6.99 (m, *J* = 8.94 Hz, 2 H), 7.37 (m, *J* = 8.94 Hz, 2 H), 7.82 (s, 1 H), 7.89 (d, *J* = 4.81 Hz, 1 H), 8.02–8.03 (m, 1 H), 8.27 (d, *J* = 1.37 Hz, 1 H); ^13^C NMR (176 MHz, DMSO-*d*_6_) δ ppm 55.91, 106.22, 115.09, 115.61, 117.30, 119.56, 120.76, 123.68, 130.03, 132.08, 132.94, 137.74, 142.37, 144.52, 149.04, and 155.55. *m/z* (ESI-MS) [M]^+^ 336.18.

### 3.2. In Vitro CDK9 Kinase Assay

The in vitro kinase activity was measured using a CDK9 assay kit obtained from BPS Biosciences (San Diego, CA). The inhibitory effect of the tested compounds was assessed following the manufacturer’s instructions as indicated in the kit. GraphPad Prism 5.0 software was used to analyze the results. DMSO was used as a negative standard in this assay, and dinaciclib was used as a positive standard.

### 3.3. MTT Cytotoxicity Assay

Cells were obtained from American Type Culture Collection. The cell culture DMEM was obtained from Life Technologies and Invitrogen and supplied with 10% FBS from Hyclone, 1% penicillin-streptomycin, and 10 µg/mL insulin from Sigma-Aldrich.

The MTT assay was used to monitor the in vitro cytotoxicity of our compounds. The cells were treated with serial concentrations of the compounds to be tested, which ranged from 0.1–10 µM at 37 °C for 48 h. Then, the cells were incubated with 10% *v*/*v* reconstituted MTT at 37 °C for 3 h. The multiwell plates were then read using Wallac Victor2 1420 multilabel counter, and the absorbance was measured at a wavelength of 450 (ex) and 590 nm (em).

### 3.4. Antiviral Assay

The cytopathic inhibition effect was used to assess the antiviral and cytotoxicity of the compounds using a method described by Choi et al. (2009) [49]. Coronavirus 229E cells were used in this assay. The antiviral activity of the tested compounds was measured as a percentage using a method described by Pauwels et al.’s (1988) research [50]. GraphPad Prism 5.0 software was used to analyze the results. Ribavirin was used as a positive standard in this assay, and DMSO was used as a negative standard.

### 3.5. Docking Studies

In these studies, the required proteins’ crystal structures were gained from PDB (3LQ5: CDK9 costructure with CR8 ligand; 6W63: COVID-19 costructure with X77 ligand). The proteins were downloaded as PDB files and prepared using Discovery Studio by keeping one subunit and removing the water and other ligands. The prepared protein was then saved in PDB format and converted to PDBQT format using AutoDock. The required inhibitors (compounds **4a**, **3a**, **2c**, and **3c**) were prepared by ChemDraw Ultra 14.0 and then saved as a PDB file and used as ligands in the docking studies. The docking studies were performed using AutoDock Vina implemented in PyRx. The analysis of the obtained docking results was conducted using Discovery Studio.

## 4. Conclusions

In conclusion, new imadazopyrazines as first-in-class CDK9 inhibitors were synthesized and biologically evaluated for their anticancer and antiviral activity. The new derivatives were assessed in vitro against isolated CDK9, and the data of this assay showed that our compounds demonstrated a good CDK9 inhibition effect with an IC_50_ of 0.18–1.78 µM. In the MTT cytotoxicity assay against the MCF7, HCT116, and K652 cancer cell lines, our compounds demonstrated good antiproliferative effects, with an average IC_50_ in the three cell lines ranging from 10.65 to 143.79 µM. In addition, the results of this assay showed a correlation between the antiproliferative effects of the inhibitors and their inhibition of CDK9, which suggests the CDK9 inhibition as a mechanistic pathway for their anticancer effects. The physiochemical and pharmacokinetic parameters of the new agents were predicated in silico, and they exhibited reasonable drug-likeness properties. The compounds with the most promising data were further assessed for their antiviral activity against human Coronavirus 229E, and the most potent agent showed a good inhibitory effect with an IC_50_ of 63.28 µM and a selectivity index of 4.8. This data was supported by docking studies with a COVID-19 main protease, which showed a high binding affinity.

## Data Availability

Data is contained within the article.

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
