# Peer review of "New Imadazopyrazines with CDK9 Inhibitory Activity as Anticancer and Antiviral: Synthesis, In Silico, and In Vitro Evaluation Approaches"

_pharmaceuticals, 2023, doi:10.3390/ph16071018_

Round 1
Reviewer 1 Report
Manuscript entitled, “New imadazopyrazines with CDK9 inhibitory activity as anti-2 cancer and antiviral; synthesis, in silico and in vitro evaluation 3 approaches” need revision before to be accepted.
1. The abstract section should be more informative; authors are advised to reorganize with brief information regarding the problem and add significant values of your results.
2. Add some more updated references in the introduction section.
3. Authors are advised to improve English; several grammatical errors are there in the manuscript.
4. Figure 1 should be clearer.
5. In Cytotoxicity assessment is there any reason with highest 10 µM?
6. Authors are advised to rearrange the abstract to make a proper coherence in the conclusion to be more interesting and more elaborative and informative.
7. In table the Docking score is almost similar is there any reason?
8. Line 206, the standard ribavirin (well-established antiviral 206 agents) was used as control in this experiment, is not clear which type of control?
9. Line 219” This suggesting the inhibition of protases enzyme as a mechanistic path-219 way of observed anti-coronaviral activity of 4a in in vivo assay? How can you claim?
10. In line 233” The main H-bonding 232 between Gly143 and Glu166 amino acid residues of the receptor and the cocrystalized X77 233 remove residues or aminoacids.
11. Authors are advised to provide high resolution figure part (interaction) presented with various colors.
12. Overall, the manuscript is good, and the information’s are clear therefore, recommended for acceptance.
Extensive editing of English language required
Reviewer 2 Report
The MS entitled “New imadazopyrazines with CDK9 inhibitory activity as anti-2 cancer and antiviral; synthesis, in silico and in vitro evaluation 3 approaches” is on an interesting topic but is not suitable for publication in the present form. It needs improvement considering the following comments:
1. The English is very poor throughout the MS and needs significant improvement. Some of the mistakes are mentioned below:
a) CDK9 is a member of this family that in collaborating with cyclin… should be replaced by CDK9 is a member of this family that in collaboration with cyclin
b) There is a satisfactory evident that support the role of CDK9 should be replaced by There is a satisfactory evidence that support the role of CDK9
c) the big pharmaceutical companies were developed several scaffolds as CDK9 inhibitors should be replaced by, the big pharmaceutical companies developed several scaffolds as CDK9 inhibitors
d) Several of which were succeed to progress should be rephrase.
e) table 1 legend should be corrected
f) table 2 legend should be corrected
g) by our team was further developed should be rephrase.
h) the activity of the new compounds will be evaluated in MTT assay should be replace by the activity of the new compounds has been evaluated in MTT assay
i) the antiviral effect of the agents will be assessed should be replaced by the antiviral effect of the agents have been assessed
2. The authors should have mentioned the recent reviews on CDK inhibitors in the introduction. Some of them are given below
a) CDK inhibitors in cancer therapy, an overview of recent development Am J Cancer Res. 2021; 11(5): 1913–1935.
b) Naturally Sourced CDK Inhibitors and Current Trends in Structure-Based Synthetic Anticancer Drug Design by Crystallography, Anti-Cancer Agents in Medicinal Chemistry, 2022, 22(3), pp. 485–498.
3. The author should give rational behind the substitutions, particularly for R.
4. The SAR studies have not been well discussed, the authors may co-relate the IC50 values with physicochemical parameters as well as CDK inhibitory activities with cytotoxicity activities against different cell line.
5. The docking scores/interactions should also be co-related with CDK inhibitory activities so as to explain the variation in activity vis-à-vis interactions and the active site of enzyme. The authors may look into recent publications in this reference.
a) Molecular docking-based interactions in QSAR studies on Mycobacterium tuberculosis ATP synthase inhibitors, SAR and QSAR in Environmental Research 33 (4), 289-305.
6. The authors have discussed the Docking studies of compound 4a with COVID-19 main protease in isolation and it would have been better to compare their results with the similar studies done earlier for example:
a) The antiviral and antimalarial drug repurposing in quest of chemotherapeutics to combat COVID-19 utilizing structure-based molecular docking
Combinatorial Chemistry & High Throughput Screening 24 (7), 1055-1068, 2021,
b) Exploring spike protein as potential target of novel coronavirus and to inhibit the viability utilizing natural agents Current Drug Targets 22 (17), 2006-2020, 2021.
The quality of English language is poor and needs significant improvement.
Round 2
Reviewer 1 Report
accepted
Reviewer 2 Report
The revised MS has been significantly improved and may be accepted for publication.
Minor corrections are still needed in English.